# All trans-retinoic acid modulates hyperoxia-induced suppression of NF-kB-dependent Wnt signaling in alveolar A549 epithelial cells

**Nikolaos Tsotakos**[1][◉], **Imtiaz Ahmed**[2][◉][¤], **Todd M. Umstead**[2], **Yuka Imamura**[3,4], **Eric Yau**[2], **Patricia Silveyra**[2,5,6], **Zissis C. Chroneos**[2,4,7] *

**1** School of Science, Engineering, and Technology, Penn State Harrisburg, Middletown, Pennsylvania, United States, **2** Department of Pediatrics, Division of Neonatal-Perinatal Medicine, Pulmonary Immunology and Physiology Laboratory, Pennsylvania State University College of Medicine, Hershey, Pennsylvania, United States of America, **3** Departments of Pharmacology and Biochemistry and Molecular Biology, Pennsylvania State University College of Medicine, Hershey, Pennsylvania, United States of America, **4** Institute of Personalized Medicine, Pennsylvania State University College of Medicine, Hershey, Pennsylvania, United States of America, **5** Department of Environmental and Occupational Health, School of Public Health, Indiana University Bloomington, Bloomington, Indiana, United States of America, **6** Division of Pulmonary, Critical Care, Sleep & Occupational Medicine, Department of Medicine, Indiana University School of Medicine, Indianapolis, Indiana, United States of America, **7** Department of Microbiology and Immunology, Pennsylvania State University College of Medicine, Hershey, Pennsylvania, United States of America

◉ These authors contributed equally to this work.
¤ Current address: Stormont Vail Health, Topeka, Kansas, United States of America
* zchroneos@pennstatehealth.psu.edu

**Data Availability Statement:** https://www.ncbi.nlm.nih.gov/geo/query/acc.cgi?acc=gse202458.

## Abstract

### Introduction

Despite recent advances in perinatal medicine, bronchopulmonary dysplasia (BPD) remains the most common complication of preterm birth. Inflammation, the main cause for BPD, results in arrested alveolarization. All trans-retinoic acid (ATRA), the active metabolite of Vitamin A, facilitates recovery from hyperoxia induced cell damage. The mechanisms involved in this response, and the genes activated, however, are poorly understood. In this study, we investigated the mechanisms of action of ATRA in human lung epithelial cells exposed to hyperoxia. We hypothesized that ATRA reduces hyperoxia-induced inflammatory responses in A549 alveolar epithelial cells.

### Methods

A549 cells were exposed to hyperoxia with or without treatment with ATRA, followed by RNA-seq analysis.

### Results

Transcriptomic analysis of A549 cells revealed ~2,000 differentially expressed genes with a higher than 2-fold change. Treatment of cells with ATRA alleviated some of the hyperoxia-induced changes, including Wnt signaling, cell adhesion and cytochrome P450 genes, partially through NF-κB signaling.

**Funding:** This work was funded by a Pediatrics Research Discovery and Education Fund awarded to IA through the Neonatal-Perinatal Fellowship program, Department of Pediatrics, Penn State College of Medicine. The funders had no role in study design, data collection and analysis, decision to publish, or preparation of the manuscript.

**Competing interests:** The authors declare that no competing interests exist.

## Discussion/Conclusion

Our findings support the idea that ATRA supplementation may decrease hyperoxia-induced disruption of the neonatal respiratory epithelium and alleviate development of BPD.

## Introduction

Bronchopulmonary Dysplasia (BPD) is a major cause of morbidity and mortality in preterm infants, affecting 10,000–15,000 infants per year in the United States [1–3]. Despite recent advances in medicine, BPD remains the most common complication of preterm birth [1, 2].

Lung morphogenesis in preterm infants with BPD arrests during the saccular phase of lung development when terminal airways branch and expand into alveolar ducts. Arrested alveolarization leads to decreased saccular branching and simplified airways, resulting in dilated terminal airways with fewer branches and lower number of alveoli. The lower number of alveoli reduces the surface area available for gas exchange [2]. Despite our evolving understanding of the pathogenesis of BPD, the normal signals for alveolar septation and the mechanisms underlying the disease pathobiology remain unknown [2, 4].

Inflammation is one of the key factors that results in disruption of normal lung development [2]. In preterm neonates, inflammatory processes result from several postnatal factors including mechanical ventilation, exposure to inhaled oxygen and other sources of oxidants such as parenteral nutrition, sepsis, and patent ductus arteriosus. Studies in mice, rats, and baboons have shown that sustained hyperoxia inhibits alveolar formation [5–7]. More recent studies have shown that inflammation can disrupt normal human development in extremely preterm infants [1–3, 8]. At the molecular level, this represents an interference between the inflammatory signaling pathways and normal developmental mechanisms [1, 2, 4].

Due to the multi-factorial causes for inflammation, an effective cure for BPD remains elusive [1, 2, 8]. Current therapeutics for BPD involve ventilation management, steroid treatment, and administration of various agents, such as pulmonary surfactant, caffeine, vitamin A, nitric oxide, and stem cells [1, 8]. Of all the therapeutic interventions for BPD, only caffeine and vitamin A have been shown to be beneficial in preventing BPD [3, 8, 9]. Although much is known about the mechanism of action of caffeine, information regarding the mechanism of action of vitamin A remains limited. Current evidence suggests that all trans-retinoic acid (ATRA), one of the active metabolites of vitamin A, facilitates recovery from hyperoxia-induced cell damage [10, 11]. ATRA is a non-selective retinoid that binds to all subtypes of retinoic acid receptors (RARα, RARβ, and RARγ) [12]. Yet, the mechanisms involved in this response and the downstream genes activated upon supplementation with ATRA are poorly understood. In A549 cells, a cell line used to model lung epithelia, ATRA may activate the ERK and Akt signaling pathways in a transcription-independent manner, with pro-survival effects [13, 14], while it can regulate RAR transcript levels [15]. Additionally, a 10:1 molar combination of vitamin A and ATRA attenuates hyperoxia-induced alterations in lung function of newborn mice, presumably by modulating the expression of proinflammatory molecules, such as MIP2α and IFNγ [16]. In the present study, we investigated the mechanism of action of ATRA in A549 cells exposed to hyperoxia by analysis of the transcriptome. We hypothesized that ATRA attenuates hyperoxia-induced inflammation in A549 alveolar epithelial cells.

## Materials and methods

### Reagents

All-trans retinoic acid (ATRA) was obtained from Sigma-Aldrich (St. Louis, MO). A $10^{-4}$ mM ATRA stock solution was made using 100% ethanol (ETOH) and stored at -80˚C in amber glass vials. This was further diluted in ETOH to achieve the desired concentration for the different treatment groups. Control cells were treated with ETOH vehicle.

### Cell culture and exposure of cells to hyperoxia

We utilized A549 human lung epithelial cells (ATCC CCL-185), cultured in DMEM media (Mediatech, Inc., Manassas, VA) containing 10% fetal bovine serum (Atlanta Biologicals, Inc., Flowery Branch, GA), and 1% penicillin/streptomycin (Mediatech, Corning, NY). Cells were used within a range of two passages for all experiments conducted, after being thawed from liquid nitrogen and cultured for ~3–4 days prior to being used in experiments. For experiments, cells were sub-cultured in 6-well plates at a density of 1 x $10^5$ cells/well. Cells were subjected to hyperoxia (95% $O_2$) for 24, 48 and 72 hours, along with control cells in normoxia (21% $O_2$), in the presence or absence of $10^{-6}$, $10^{-5}$, and $10^{-4}$ mM all-trans retinoic acid (ATRA). Hyperoxia experiments were conducted in a Modular Incubator Chamber (MIC-101, Billirups-Rothenberg Inc., Del-Mar, CA) following the manufacturer's protocol. This chamber was then maintained at 37˚C in a humidified incubator.

### Cell counting

Cells were harvested to assess time- and dose-responses to treatments. Viable cells were counted in 0.4% Trypan Blue in phosphate buffered saline (ThermoFisher Scientific) to exclude trypan blue permeable cells using a hemocytometer. Each experiment was repeated in triplicate and viable cell number expressed as the mean±SD of for each condition.

### Measure of transcriptome response

Cells were harvested and RNA was extracted at the end of the 72-hour time period. Total RNA was purified using RNA-Bee (Tel-Test Inc., Friendswood, TX) and submitted for RNAseq analysis at Penn State's Functional Genomics Core. Stranded indexed cDNA libraries were prepared with the NEXTFLEX® Rapid Directional RNA-Seq Library Prep Kit (Perkin Elmer, Waltham, MA) and sequenced at Penn State's Functional Genomics Core.

### Bioinformatics analysis of transcriptomic data

Quality of reads obtained from each sample was assessed using FastQC software. Adapter sequences were trimmed, and low length reads (<50nt) were discarded with Trimmomatic (v.0.38) [17]. Pseudoalignment of reads to the human transcriptome was performed with Kallisto, followed by quantification with Sleuth [18, 19]. Alignment of the filtered reads to the human genome (hg38) was performed using hisat2 and features were counted with htseq [20, 21]. Genes that were differentially expressed between groups were acquired with DESeq2 [22]. Gene Ontology and pathway analysis was performed with the use of the topGO and gage R packages [23, 24]. Transcriptome results were analyzed by IPA ingenuity pathway analysis (QIAGEN Bioinformatics, Redwood City CA). Significant gene expression changes between treatments were determined with an FDR of q<0.05, and gene networks were obtained using IPA and/or DAVID. Pathway inhibition or activation was assessed by z-score standard deviation analysis of log2 ratio of normalized data.

### Statistical analysis

Results were reported as means +/- standard deviation (SD). Data was analyzed using mixed effects 2-way ANOVA (the main effects were oxygen exposure and vitamin A concentration) followed by Tukey's post-test for comparison against control conditions using Graph Pad Prism 9.4 (Graph Pad Software, Inc., San Diego, CA). Significance was assigned for $p < 0.05$.

### Statement of ethics

Ethics approval is not required. This study does not involve animal or human subjects.

## Results

### Impact of ATRA concentration on cell number kinetics under normoxic conditions

The number of A549 cells (Fig 1) increased 3.5-fold by 72-hours of culture following a 24 to 48-hr lag phase (Fig 1A). The presence of $10^{-5}$ or $10^{-6}$ mM ATRA shortened the lag phase period resulting in 5.7–6.9-fold increase in cell number by 72-hours of culture, whereas the $10^{-4}$ mM concentration of ATRA did not affect cell number after 48-hrs compared to control (Fig 1A). Cell viability was estimated by cell counts at normoxia and following 24, 48 and 72 hours of hyperoxia in the presence of $10^{-6}$, $10^{-5}$ and $10^{-4}$ mM ATRA, or vehicle (control). Despite no significant difference in cellular viability after 24 hours, hyperoxia significantly reduced the number of viable A549 cells at 48 and 72 hours when compared to controls in normoxia (Fig 1).

### Impact of ATRA and hyperoxia on A549 cell number kinetics

Hyperoxia led to transient decline in A549 cell number between 24 and 48-hrs after culture, followed by recovery in cell number by 72-hours, indicating adaptation to hyperoxic conditions (Fig 1B). The presence of $10^{-5}$ mM ATRA reduced cell number loss and both $10^{-5}$ and $10^{-6}$ mM ATRA enhanced A549 cell number at 72-hours compared to controls, whereas the $10^{-4}$ mM ATRA had no effect (Fig 1B). Two-way ANOVA using ATRA concentration and oxygen exposure as the potentially contributing factors revealed all main and interaction effects to be significant, with the exception of the effect of oxygen on cell counts at the 24-hour

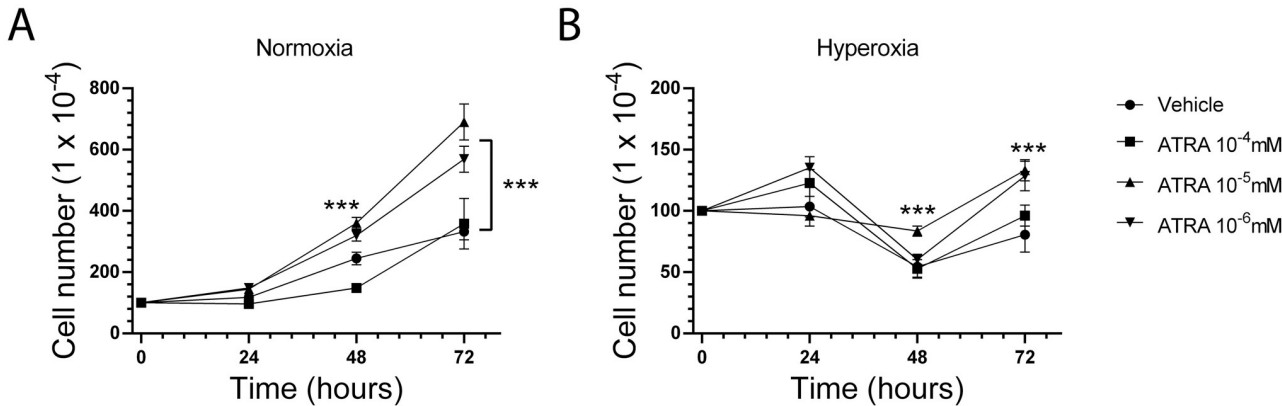

**Fig 1. Differential impact of ATRA concentration on A549 cell number kinetics.** The human A549 lung epithelial cell line was cultured under (A) normoxic (21% $O_2$) or (B) hyperoxic (95% $O_2$) conditions in the absence or presence of $10^{-4}$, $10^{-5}$ and $10^{-6}$ mM ATRA. Cells were harvested and enumerated using a hemocytometer at 24, 48, or 72hours of culture. All experiments were performed in triplicate. ***$p < 0.001$ compared to vehicle at the same time point and oxygen treatment.

time point (S1–S3 Tables). Hyperoxia suppressed the number of mRNA transcripts of the Ki67 proliferation marker at all time points (S1 Fig). The presence of $10^{-5}$ mM ATRA led to a relative increase in Ki67 mRNA in both normoxic and hyperoxic conditions, suggesting that ATRA enhances the proliferative potential of A549 cells. Changes in other markers of proliferation (PCNA and MCM2) followed the same trends, although not as prominent as Ki67 (S1 Fig).

## Unbiased transcriptome analysis of hyperoxia-exposed A549 human lung epithelial cells

We performed whole transcriptome analysis to explore gene regulatory pathways. The principal component analysis (PCA) of the RNA-seq data revealed tight clustering of all samples based on their $O_2$ exposure (normoxia or hypoxia), indicating high reproducibility of gene expression for these treatments (Fig 2A). Unsupervised clustering based on Euclidean distances separated the samples based on whether they were exposed to normoxia or induced hyperoxia (Fig 2B). A549 cells have been known to differentiate into alveolar type II cells, if they remain in culture for a prolonged period of time [25]. Terminal differentiation to alveolar type II epithelial cells was not evident in our short-term culture as indicated by the lack of surfactant protein mRNA transcripts. The presence of ATRA, however, induced low level expression of surfactant proteins, suggesting that ATRA may induce differentiation after long-term exposure (S2 Fig). We then performed differential gene expression (DGE) analysis with the hisat2-htseq-DESeq2 pipeline for a pairwise comparison of the two treatments. There were 4,541 DE genes with adjusted p-value < 0.05. Filtering these for |log2FC| >1 resulted in a list of 1,711 DE genes, of which 800 were upregulated and 911 were downregulated in the cells

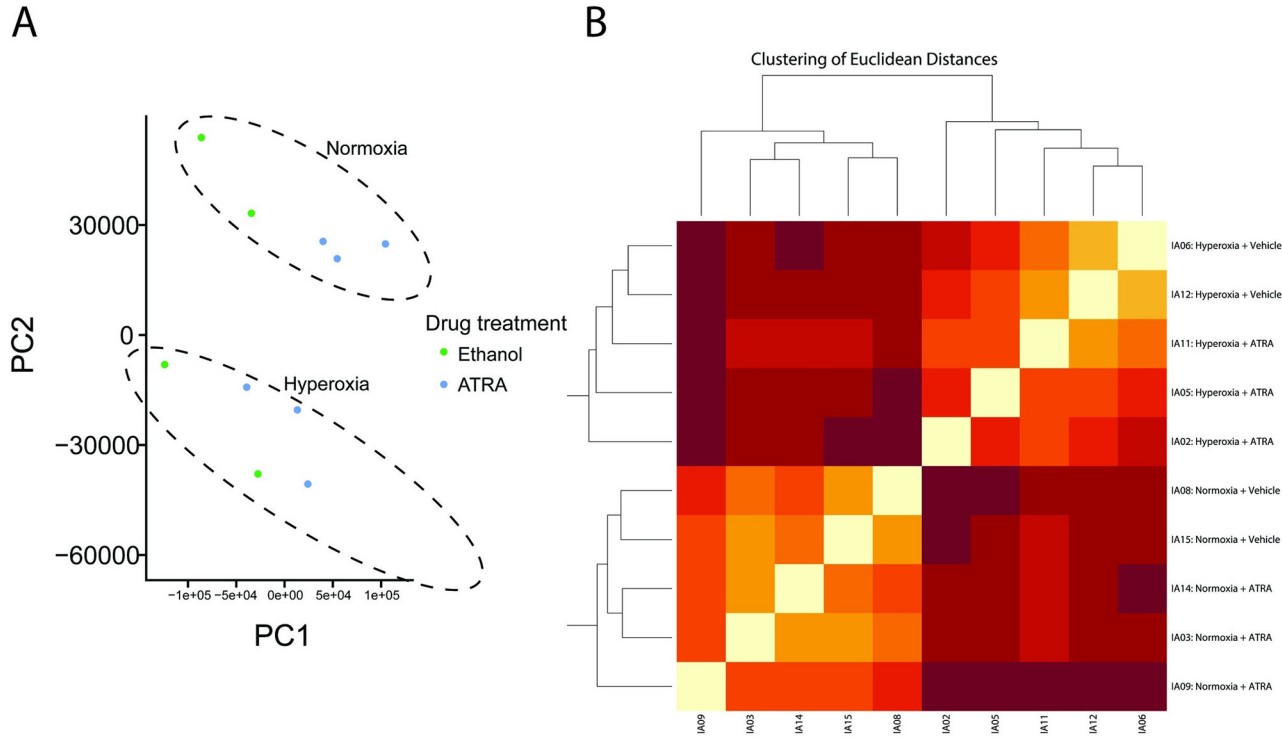

**Fig 2. Transcriptome responses of A549 cells to hyperoxia.** (A) Principal component analysis of all samples (n = 10); (B) Euclidean distances among samples.

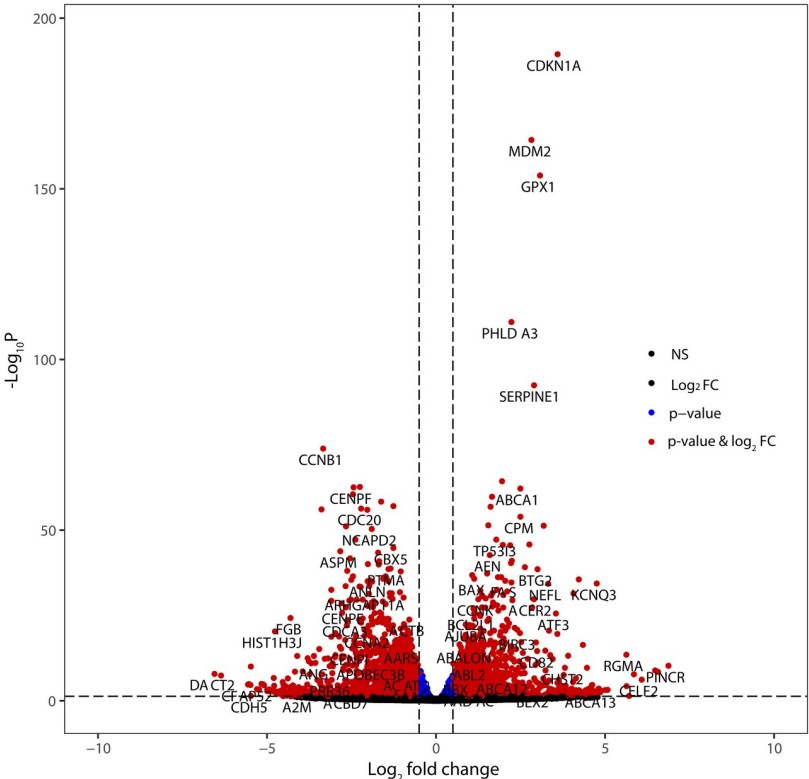

**Fig 3. Bioinformatics analysis of hyperoxia-exposed A549 cells.** Volcano plot of the differentially expressed genes. Blue dots indicate significant p-value only, while red dots indicate significant p-value and fold change.

under hyperoxia (Fig 3). The list of the genes whose expression showed the highest fold change (both upregulated and downregulated) can be seen in Table 1. We then analyzed these genes with topGO to categorize the DEGs by gene ontology (GO) terms. The top 20 biological processes associated with the DE genes by gene ontology analysis are shown in S4 Table.

## Differential gene expression in ATRA-treated A549 exposed to hyperoxia

Using the same pipeline as described above, we compared ATRA-treated A549 cells exposed to hyperoxia compared to ATRA vehicle. This approach identified 733 significantly DE genes (adjusted p-value < .05). Of these, 694 are protein coding genes, 11 were lncRNAs, and the rest were either small RNAs or pseudogenes. The genes with the highest fold change are shown in Table 2. KEGG analysis of this gene dataset with GAGE resulted indicated that the pathways associated with ATRA treatment are associated with adherens junctions (KEGG ID: hsa04520), ECM-receptor interactions (hsa04512), and $Ca^{2+}$ and Wnt signaling (hsa04020 and hsa04310, respectively), among others (Fig 4 and S5 Table).

Filtering the DE genes for a minimum 2-fold change resulted in 140 DE genes (all protein coding), of which 116 were upregulated and 24 were downregulated (Fig 5A and 5B, S6 Table). The top 20 GO terms associated with these changes can be seen in S7 Table. Next, we determined whether ATRA treatment rescued the expression of any genes differentially expressed in the hyperoxia-exposed cells. Of the 800 upregulated genes, there were 11 that were rescued during treatment with ATRA. Inversely, the expression of 23 of the 911 downregulated genes was rescued by treatment with ATRA (Fig 5C).

**Table 1. Top 20 up- and down-regulated genes in A549 cells exposed to hyperoxia.**

| Upregulated in hyperoxia | | | Downregulated in hyperoxia | | |
|---|---|---|---|---|---|
| Gene Symbol | log2 FC | adjusted p-value | Gene Symbol | log2 FC | adjusted p-value |
| PINCR | 6.874166 | 1.87E-09 | DACT2 | -6.5516 | 1.24E-08 |
| HCAR3 | 6.568128 | 7.67E-08 | LOC105375151 | -6.36146 | 4.36E-08 |
| DRAXIN | 6.485795 | 3.33E-08 | LOC105370756 | -5.56523 | 1.68E-05 |
| CELF2 | 6.087206 | 1.01E-05 | LOC101928351 | -5.53619 | 1.48E-05 |
| FAM84A | 5.84974 | 3.55E-07 | CFAP52 | -5.49228 | 2.16E-05 |
| LOC100131315 | 5.63978 | 0.000504 | PIF1 | -5.48257 | 9.47E-11 |
| RGMA | 5.630156 | 1.51E-12 | LINC00284 | -5.29562 | 0.00042 |
| C10orf105 | 5.074823 | 0.00444 | LOC105372815 | -5.24861 | 0.000517 |
| FETUB | 5.00955 | 0.004676 | TRPM8 | -5.23713 | 0.000501 |
| PPP1R3G | 4.856756 | 0.018533 | HPDL | -5.16016 | 1.14E-05 |
| ARMH4 | 4.850103 | 0.004639 | LOC107984500 | -5.11214 | 1.52E-05 |
| SCN2A | 4.82479 | 0.003153 | TRS-GCT6-1 | -5.08734 | 2.71E-05 |
| CABP5 | 4.81651 | 0.001906 | LOC101928222 | -4.99945 | 0.000653 |
| ALOX5 | 4.803004 | 0.002878 | ITGB2 | -4.99012 | 0.00145 |
| KCNQ3 | 4.750091 | 1.27E-32 | DEFB1 | -4.95166 | 0.001371 |
| UCA1 | 4.723688 | 0.013616 | LOC107984763 | -4.93838 | 0.000507 |
| TRIB2 | 4.71115 | 0.033593 | TNNC1 | -4.90801 | 0.00145 |
| LOC105372352 | 4.710667 | 0.032733 | IL7 | -4.83649 | 0.001397 |
| ADRB2 | 4.668502 | 0.029264 | TRIM31 | -4.80627 | 2.34E-07 |
| ABCA13 | 4.657653 | 0.009008 | LOC107985555 | -4.78471 | 3.92E-05 |

**Table 2. Top 20 up- and down-regulated genes in hyperoxia exposed A549 cells in the presence of $10^{-}5$ mM ATRA.**

| Upregulated in hyperoxia | | | Downregulated in hyperoxia | | |
|---|---|---|---|---|---|
| Gene Symbol | log2 FC | adjusted p-value | Gene Symbol | log2 FC | adjusted p-value |
| CYP26B1 | 4.131202 | 1.05E-17 | RPS29 | -2.28161 | 1.33E-68 |
| ATOH8 | 2.302342 | 0.002083 | IGF2 | -1.95797 | 4.51E-12 |
| SFRP4 | 2.260821 | 8.02E-07 | INS-IGF2 | -1.95475 | 1.76E-11 |
| VTN | 2.241636 | 2.24E-06 | MXD1 | -1.39811 | 0.012619 |
| CEACAM6 | 2.229989 | 3.80E-21 | TFPT | -1.37662 | 0.001453 |
| ISLR | 2.122232 | 0.001524 | STARD7-AS1 | -1.29073 | 0.011424 |
| GALNT12 | 2.113117 | 3.75E-05 | HMGA2 | -1.28564 | 7.59E-08 |
| STRA6 | 1.868627 | 2.34E-38 | FRMD6 | -1.2559 | 3.40E-08 |
| DHRS3 | 1.864324 | 3.76E-32 | GALNT13 | -1.24584 | 0.006976 |
| CYP2S1 | 1.855462 | 2.05E-07 | CA8 | -1.21492 | 2.65E-06 |
| CYP24A1 | 1.842927 | 4.20E-56 | NAMPT | -1.20134 | 4.43E-18 |
| C3 | 1.79089 | 1.28E-07 | RSPO3 | -1.18388 | 1.43E-19 |
| PTGES | 1.784223 | 8.51E-09 | EREG | -1.18216 | 0.001584 |
| C1R | 1.778613 | 4.49E-13 | TLE4 | -1.15773 | 0.027043 |
| TNS4 | 1.76341 | 2.60E-06 | LAMC2 | -1.12234 | 2.12E-05 |
| RHCG | 1.731553 | 1.22E-15 | PHLDA2 | -1.08888 | 0.032753 |
| PCK2 | 1.700365 | 5.70E-05 | KYNU | -1.08124 | 1.31E-19 |
| ELF3 | 1.679172 | 1.36E-24 | GEM | -1.07954 | 0.032875 |
| TFPI2 | 1.655075 | 4.11E-33 | TMEM265 | -1.03037 | 0.015111 |
| LXN | 1.617019 | 7.57E-05 | CREM | -1.02606 | 0.015313 |

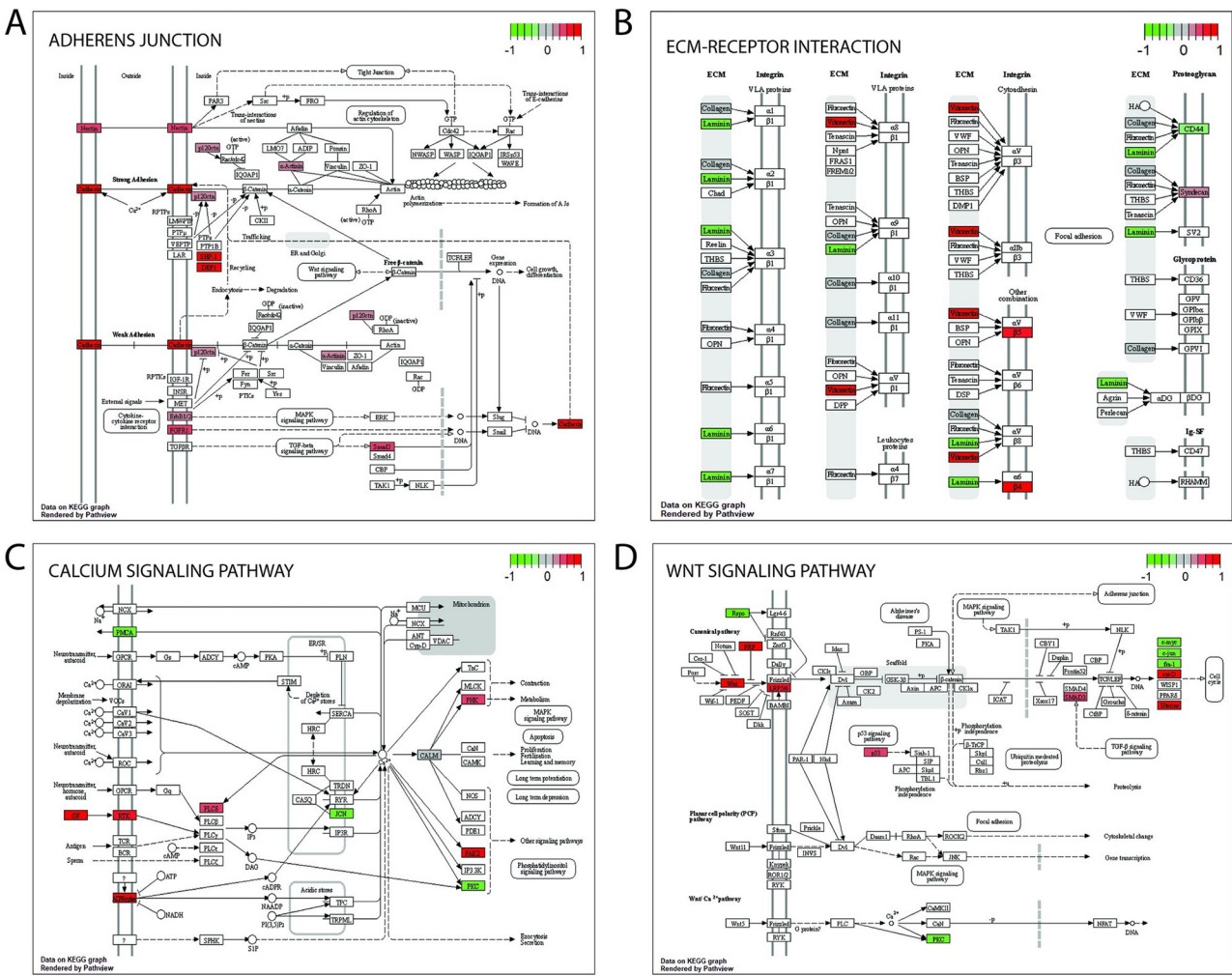

**Fig 4. Pathway analysis of the effects of ATRA treatment of hyperoxic A549 cells.** KEGG (Kyoto Encyclopedia of Genes and Genomes) analysis was performed with GAGE (Generally Applicable Gene-Set Enrichment for Pathway Analysis). The full list of pathways can be seen in S5 Table. (A) adherens junctions; (B) ECM-receptor interaction; (C) calcium signaling; (D) Wnt signaling.

## Hyperoxia causes upregulation of genes responsible for inflammation and inhibition of genes responsible for cellular proliferation

We next analyzed the RNA-seq data with the Database for Annotation, Visualization and Integrated Discovery (DAVID), as well as the Ingenuity Pathway Analysis (IPA). Upstream pathway analysis using IPA revealed significant upregulation of genes involved in the oxidative stress and innate immune response pathways in cells exposed to hyperoxia. This analysis also showed that hyperoxia enhances mRNA levels of RELB and NFκB2, which are components of the NF-κB signaling complex (S7 Table). Similarly, use of the UCSC TFBS (Transcription Factor Binding Sites) tool on DAVID revealed that the NF-κB pathway is significantly affected not only in cells exposed to hyperoxia, but also when cells are pretreated with ATRA. The change in expression levels of DE NF-κB target genes in cells pre-treated with ATRA compared to vehicle can be seen in Fig 6. Interestingly, the levels of NF-kB2 itself are partially rescued by ATRA treatment (Figs 5c and 6a). Additionally, the hyperoxia-induced perturbation of expression of several genes related to cell adhesion was reverted with ATRA treatment (Fig 6b).

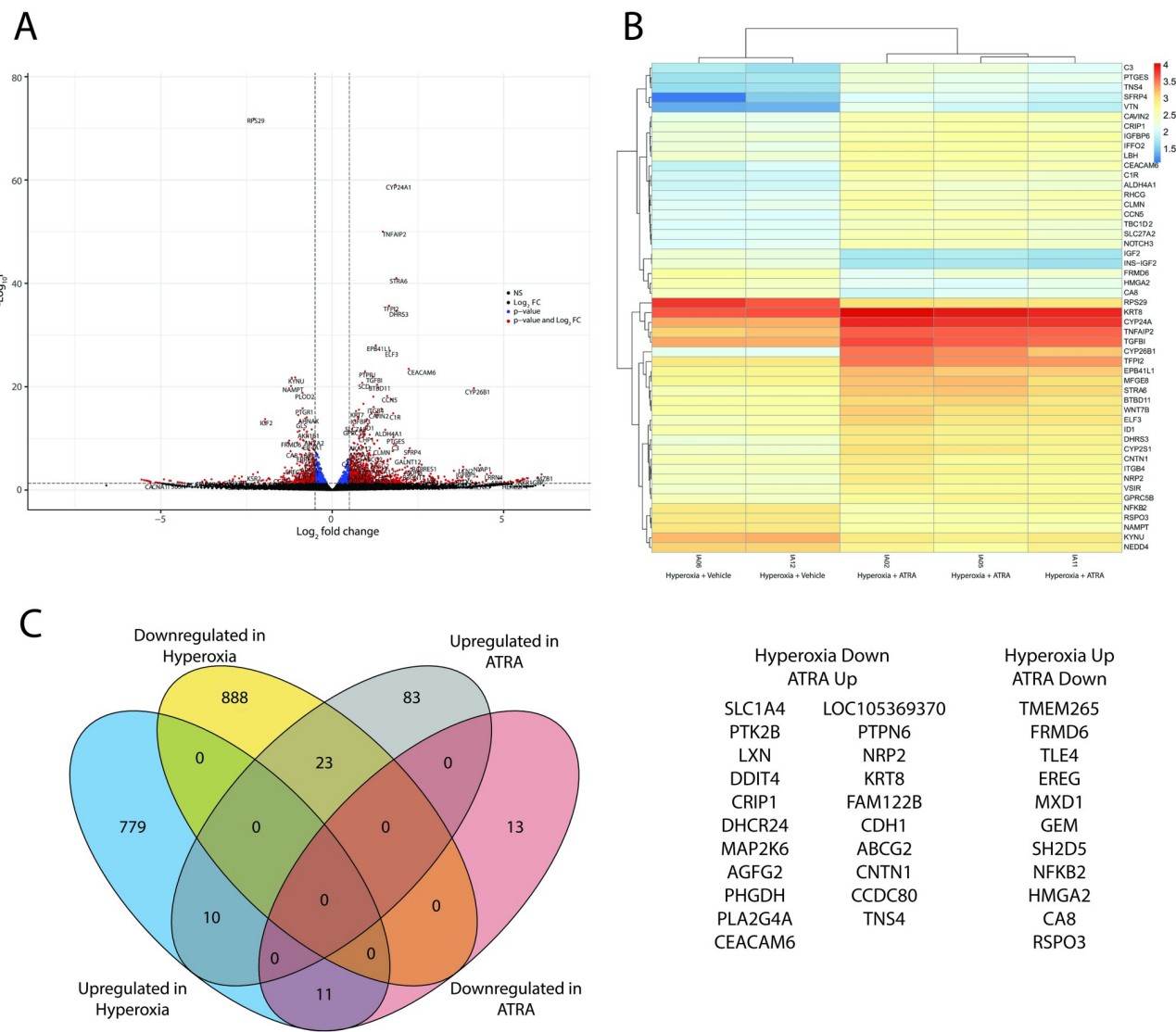

**Fig 5. Bioinformatics analysis of hyperoxic A549 cells treated with ATRA or vehicle (EtOH).** (A) Volcano plot of the differentially expressed genes. Blue dots indicate significant p-value only, while red dots indicate significant p-value and fold change; (B) Heatmap of gene expression changes of the top 50 most dysregulated genes based on adjusted p-value; (C) Venn diagram showing the dysregulated genes; the lists to the right indicate hyperoxia-induced genes that were suppressed by treatment with ATRA and hyperoxia-suppressed genes that were upregulated by treatment with ATRA.

## Discussion/Conclusion

In this study, we report that hyperoxia significantly reduces proliferation of A549 cells in culture as indicated by delayed increase in cell number over time when compared to controls in normoxia. Treatment with ATRA enhanced proliferation of A549 cells at low but not high concentration under both normoxic and hyperoxic conditions, although this effect was observed more than 48 hours after addition of ATRA in cells exposed to hyperoxia. ATRA exerts concentration-dependent effects on cellular proliferation, survival, and differentiation through genomic and non-genomic mechanisms [13, 14, 26, 27]. Here, and intermediate concentration of $10^{-5}$ mM ATRA was optimal to facilitate adaptation to hyperoxia and restore proliferation. The results of our study also highlight that hyperoxia inhibits genes responsible

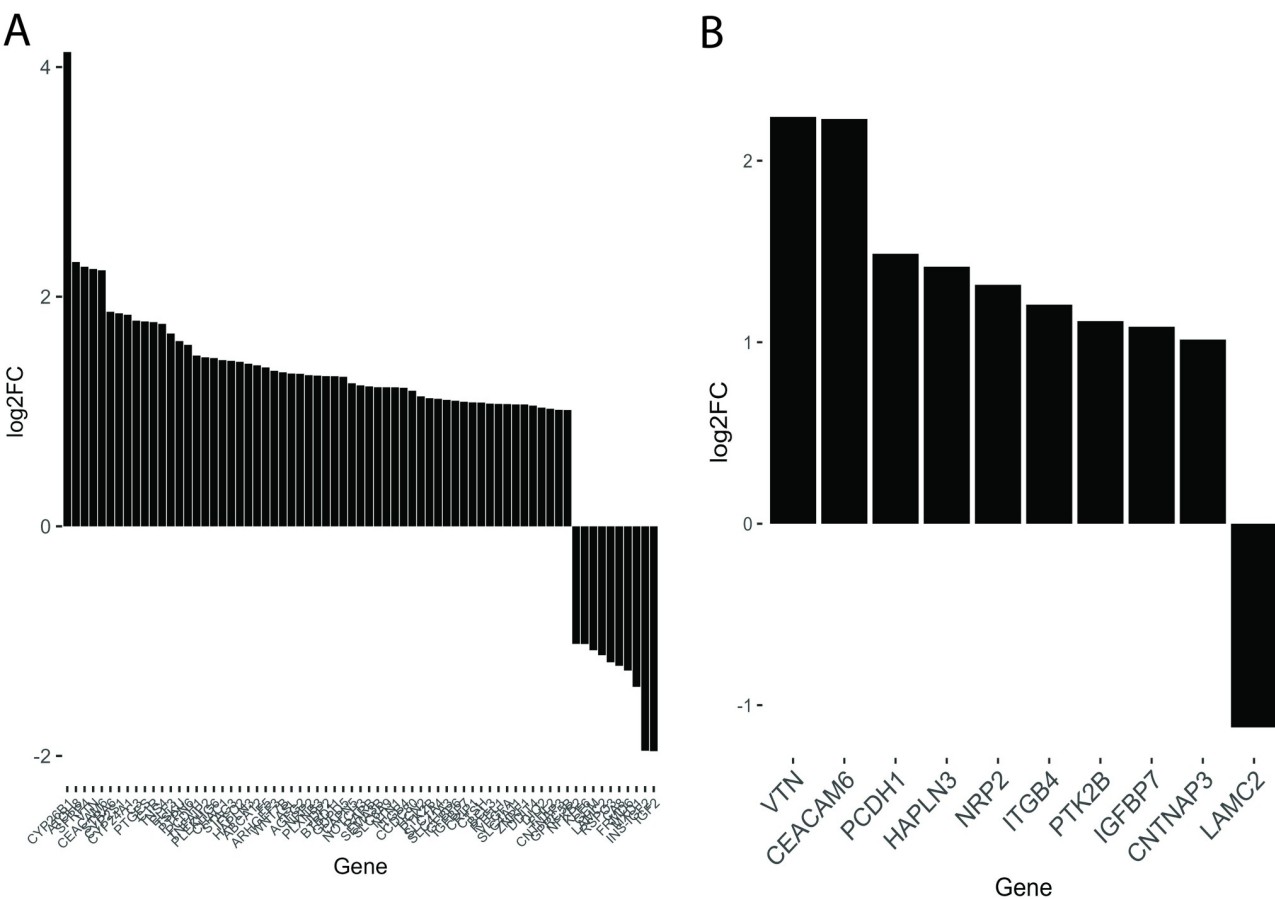

**Fig 6. Network analysis of genes rescued by ATRA.** Upstream analysis of hyperoxia-dysregulated genes whose expression was rescued by ATRA as determined by the Database for Annotation, Visualization and Integrated Discovery (DAVID). (A) Expression levels of ATRA-rescued NF-kB target genes following hyperoxia; (B) Expression levels of NF-kB target genes that are associated with cell adhesion (GO: 0007155).

for epithelial cell adhesion, such as integrin β2, as well as Wnt signaling modulators, such as DACT2 (Disheveled binding Antagonist of beta Catenin-2), consistent with the transient reduction in adherent A549 cells exposed to hyperoxia. Taken together, our findings buttress clinical observations of BPD prevention with vitamin A supplementation [2, 28] and the association between hyperoxia and inflammation [2, 10].

Analysis of the transcriptome gene response revealed that hyperoxia exposure leads to a dysregulation of the Wnt signaling pathway and adhesion properties of the cells, such as adherens junctions and interactions with the ECM, among others. Precise activation of the Wnt/β-catenin signaling pathway is critical for normal development of the lungs [29, 30]. In early lung formation, retinoic acid locally activates the Wnt pathway via modulation of Dkk1, thus affecting interactions with other pathways, like the ones involving Tgf-β and Fgf10 [31]. Hyperoxia has been shown to induce mesenchymal Wnt5A, which can affect lung morphogenesis in 3D tissue cultures [32]. In our experiments, ATRA supplementation partially rescued these perturbed phenotypes, likely through suppression of the hyperoxia-induced activation of the NF-κB signaling pathway, although additional studies are needed to test this idea.

NF-κB is an important regulator of cellular responses and is present in the cell in an inactive state (rapid-acting primary transcription factors) [33]. Signal-induced degradation of IκB

proteins (inhibitors of κB) brought about by harmful cellular stimuli (i.e., LPS, reactive oxygen species, TNFα) results in activation of NFκB. With the degradation of IκB, the NFκB complex translocates into the nucleus where it can induce expression of specific genes which brings about physiological responses such as inflammation, cell survival responses and cellular proliferation [33].

Notably, the expression of three NF-κB target genes, namely WNT7B, RSPO3, and SFRP4, that are associated with the Wnt signaling pathway was rescued by ATRA treatment of hyperoxia-challenged A549 cells. Of these, Wnt7b is a member of the Wnt gene family, and R-Spodin 3 is an activator of the canonical Wnt signaling pathway by binding to the LGR4-6 receptors. It can also act as a regulator of both canonical and non-canonical Wnt signaling by inhibiting ZNRF3 [34]. SFRP4 (Secreted Frizzled-Related Protein 4) is a soluble protein that contains a cysteine-rich domain homologous to the putative Wnt-binding site of Frizzled proteins. It is secreted by its producing cell, thereby modulating Wnt signaling.

Importantly, several members of the cytochrome P450 superfamily were upregulated following ATRA treatment of A549 cells exposed to hyperoxia, including CYP26B1, which had the highest fold change, CYP24A1, and CYP2S1. Of these, CYP26B1 is expressed in the distal tip of the forming limb bud in developing mice. Cyp26b1$^{-/-}$ mice manifest with severe limb malformations and die after birth during respiratory illness [35]. These mice exhibit lung defects, most notably decreased distal airspaces and hypercellular tissues with decreased air spaces [36]. The gene product is an RA-catabolizing enzyme, but it may have roles in lung morphogenesis beyond RA regulation, including the regulation of alveolar epithelial type I and type II cells [36]. Cyp24a1, a monooxygenase that helps degrade 1,25-dihydrovitamin D3, may also play a role in the proliferation of alveolar epithelial type II cells in the developing lung, as vitamin D deficiency has been associated with lower alveolar type II cell numbers and the prevalence of BPD in preterm infants [37, 38]. CYP2S1 is not as well studied as the other cytochrome P450 genes identified here. However, depletion of CYP2S1 in bronchial epithelial cells lead to changes in cell adhesion, metabolism of lipids and arachidonic acid, as well as RA metabolism [39]. Loss of Cyp2s1 seems to divert the RA flux into PPAR signaling rather than metabolic inactivation, given the increase of FABP5 and the lack of oxidizing enzymes in these cells. In our system, the increased Cyp2s1 may simply be a direct response to the increased levels of retinoic acid due to the ATRA treatment.

There are several limitations of this study, of which the major one is that the model is an *in vitro* adult cell line, derived from adult tissue, thus the results do not easily allow a strong conclusion in the context of neonatal BPD pathophysiology. The observed transcriptome response also remains to be validated at the protein level and in an animal model. In this context, models of exposure of mice to hyperoxia that mimic BPD phenotypes [40] would provide better insights in terms of BPD pathophysiology. Furthermore, additional studies are needed to assess the effects of hyperoxia and ATRA on cell-cycle entry and cellular proliferation at molecular level.

Current therapies for BPD continue to remain largely empirical given the multi-factorial etiology of BPD [2, 3, 8]. Strategies to limit the use of mechanical ventilation to prevent lung injury remain limited in the face of poor oxygenation and inadequate ventilation seen with most preterm infants during transition to extrauterine life [1, 2]. The pronounced morbidity and mortality associated with malformation or destruction of alveoli underscores a pressing need for new therapeutic concepts. Disease-modifying treatments, with the ability to halt or reverse the structural damage associated with BPD, are lacking [4]. Re-induction of alveolarization in diseased lungs is a new and exciting concept in a regenerative medicine approach to manage pulmonary diseases that are characterized by an absence of alveoli [41]. Improved understanding of the mechanisms involved in alveolarization would allow for newer

therapeutic approaches towards preventing and treating arrested alveolarization [1]. Wnt signaling-targeting therapies may be of interest to improve the regeneration of epithelial cells and as a potential future target for therapeutic intervention in our quest for a cure for BPD.

In conclusion, we show that human lung epithelial cells (A549) have decreased cellular proliferation when exposed to hyperoxia *in vitro*. This inhibition of cellular proliferation occurs due to inflammation secondary to hyperoxia. ATRA supplementation caused ablation of the inflammatory response, with partial restoration of cellular proliferation. Functional studies will be needed in animal studies and *in vitro* experiments to better understand the molecular effects of hyperoxia in lung development.

## Supporting information

**S1 Table. ANOVA summary table for cell numbers at the 24-hour time point.** (DOCX)

**S2 Table. ANOVA summary table for cell numbers at the 48-hour time point.** (DOCX)

**S3 Table. ANOVA summary table for cell numbers at the 72-hour time point.** (DOCX)

**S4 Table. Gene ontology analysis of A549 cells exposed to hyperoxia.** Following RNA-seq analysis of A549 cells treated with hyperoxia, enrichment analysis was performed with the topGO package. The top 20 terms associated with biological processes are listed. (DOCX)

**S5 Table. Pathway analysis of hyperoxia-exposed A549 cells pre-treated with ATRA.** KEGG pathways associated with the differentially expressed genes of hyperoxia-exposed cells treated with ATRA, as generated by GAGE. (CSV)

**S6 Table. List of DEG between hyperoxia-exposed A549 cells treated with ATRA and ethanol-treated controls.** A549 cells were exposed to hyperoxia in the presence or absence of ATRA, as described in Materials and Methods. RNA was isolated and sequenced, followed by analysis of the differentially expressed genes with DESeq2. (CSV)

**S7 Table. Gene ontology analysis of hyperoxia-exposed A549 cells treated with ATRA.** Following RNA-seq analysis of A549 cells treated with hyperoxia, enrichment analysis was performed with the topGO package. The top 20 terms associated with biological processes are listed. (DOCX)

**S1 Fig. Normalized read counts for proliferation markers.** Following DESeq analysis of the RNA-seq results, normalized counts for transcripts of proliferation markers Ki67, PCNA, and MCM-2 were mined. (TIF)

**S2 Fig. Normalized read counts for surfactant genes.** Following DESeq analysis of the RNA-seq results, normalized counts for transcripts of surfactant genes SFTPA1, SFTPA2, SFTPB, SFTPC, and SFTPD were mined. At least two samples per condition had zero counts for each gene. (TIF)

## Acknowledgments

The authors would like to thank Susan L. DiAngelo for technical assistance and the Penn State Institute of Personalized Medicine Genome Sciences Core Facility (RRID:SCR_021123) for assistance with mRNA sequencing.

## Author Contributions

**Conceptualization:** Patricia Silveyra, Zissis C. Chroneos.

**Data curation:** Nikolaos Tsotakos, Yuka Imamura, Eric Yau, Zissis C. Chroneos.

**Formal analysis:** Nikolaos Tsotakos, Zissis C. Chroneos.

**Funding acquisition:** Imtiaz Ahmed.

**Investigation:** Nikolaos Tsotakos, Imtiaz Ahmed, Todd M. Umstead, Yuka Imamura, Patricia Silveyra.

**Methodology:** Nikolaos Tsotakos, Imtiaz Ahmed, Todd M. Umstead, Yuka Imamura, Zissis C. Chroneos.

**Project administration:** Zissis C. Chroneos.

**Resources:** Zissis C. Chroneos.

**Supervision:** Zissis C. Chroneos.

**Validation:** Zissis C. Chroneos.

**Visualization:** Imtiaz Ahmed, Zissis C. Chroneos.

**Writing – original draft:** Nikolaos Tsotakos, Imtiaz Ahmed.

**Writing – review & editing:** Todd M. Umstead, Yuka Imamura, Eric Yau, Patricia Silveyra, Zissis C. Chroneos.

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
