## [Decision Letter · Decision Letter 0]

1 Jun 2022

PONE-D-22-12782All trans-retinoic acid modulates hyperoxia-induced suppression of NF-kB-dependent Wnt signaling in alveolar A549 epithelial cellsPLOS ONE

Dear Dr. Chroneos,

Thank you for submitting your manuscript to PLOS ONE. After careful consideration, we feel that it has merit but does not fully meet PLOS ONE’s publication criteria as it currently stands. Therefore, we invite you to submit a revised version of the manuscript that addresses the points raised during the review process.

We look forward to receiving your revised manuscript.

Kind regards,

Saverio Bellusci

Academic Editor

PLOS ONE

Journal Requirements:

Reviewers' comments:

Reviewer's Responses to Questions

**Comments to the Author**

1. Is the manuscript technically sound, and do the data support the conclusions?

Reviewer #1: Yes

Reviewer #2: Partly

2. Has the statistical analysis been performed appropriately and rigorously? 

Reviewer #1: Yes

Reviewer #2: I Don't Know

3. Have the authors made all data underlying the findings in their manuscript fully available?

Reviewer #1: Yes

Reviewer #2: No

4. Is the manuscript presented in an intelligible fashion and written in standard English?

Reviewer #1: Yes

Reviewer #2: Yes

5. Review Comments to the Author

Reviewer #1: General comments:

Bronchopulmonary dysplasia (BPD) is a common chronic lung disease in preterm infants and leads to long-term morbidity and mortality. The pathogenesis of BPD is multifactorial and occurs most commonly in preterm infants exposed to mechanical ventilation and oxygen therapy, intrauterine growth restriction, and prenatal and postnatal infections. This contributes to pulmonary growth arrest leading to alveolar simplification. Currently, there is no curative therapy.

In the present study, the authors investigated the effect of all trans-retinoic acid ATRA on proliferation and transciptomic changes in A549 cells exposed to hyperoxia.

They showed that hyperoxia-induced decrease of proliferation in A549 cells can be rescued by ATRA.

Further, the results revealed hyperoxia-induced upregulation of genes responsible for inflammation and inhibition of genes responsible for cellular proliferation. ATRA pretreatment is able to reverse some of these detrimental changes on gene level.

Major comments:

1. Trypan blue is staining dead cells. It is not clear to me how proliferation can be judge by Trypan blue staining. Please claarify. It would be better to stain for Ki67 and do flow cytometry.

2. Detection of some regulated genes of the Wnt and NF-kB pathway by RT-qPCR would underline and strengthen the RNA-seq results, because validation of the RNA-seq results is lacking.

3. Is ATRA toxic at higher doses, alone or in combination with hyperoxia?

4. The study design does not really allow a strong conclusion in the context of BPD pathophysiology. This limitation should be emphasized.

Minor comments

1. A549 cells can differentiate in the cell culture if they are kept to long and in higher passages ( Cooper JR, Abdullatif MB, Burnett EC, Kempsell KE, Conforti F, Tolley H, Collins JE, Davies DE. Long Term Culture of the A549 Cancer Cell Line Promotes Multilamellar Body Formation and Differentiation towards an Alveolar Type II Pneumocyte Phenotype. PLoS One. 2016 Oct 28;11(10):e0164438. doi: 10.1371/journal.pone.0164438. PMID: 27792742; PMCID: PMC5085087.). How long were the A549 cells in culture? What was the passage?

2. Please name the devices and companies in the materials and methods section.

3. Were biological and/or technical triplicates performed? Please also name the n-number in the figures.

4. Please check again for spelling and spaces.

5. Please be consistent in using hyperoxia-exposed, hyperoxia-treated, hyperoxia-challenged.

6. The figures are blurry. Please improve the quality.

Reviewer #2: Dear Authors,

while the transcriptome analysis is well executed, the cellular proliferation assay showed in figure 1 should be revised.

In particular the figure partially contradicts the statement at line 143 "Pretreatment with ATRA prior to hyperoxia exposure restored cellular proliferation after 48h" since the only statistical significant result at 48 hours regards the normoxia group treated wiTH 10-5 mM ATRA.

The graphical representation of the proliferation assay could be improved, and the different comparisons between groups and times is not well specified.

From a statistical point of view the proper test for this assay should be a two way ANOVA, since the 2 variables (hyperoxia time and ATRA concentration) and the results representation would probably be more clear using a linear representation of the different treatments over time.

6. PLOS authors have the option to publish the peer review history of their article (what does this mean?). If published, this will include your full peer review and any attached files.

Reviewer #1: No

Reviewer #2: No

---

## [Author Response · Author response to Decision Letter 0]

18 Jul 2022

Point by point response to reviewers' comments:

Reviewer #1: 

General comments:

Bronchopulmonary dysplasia (BPD) is a common chronic lung disease in preterm infants and leads to long-term morbidity and mortality. The pathogenesis of BPD is multifactorial and occurs most commonly in preterm infants exposed to mechanical ventilation and oxygen therapy, intrauterine growth restriction, and prenatal and postnatal infections. This contributes to pulmonary growth arrest leading to alveolar simplification. Currently, there is no curative therapy.

In the present study, the authors investigated the effect of all trans-retinoic acid ATRA on proliferation and transciptomic changes in A549 cells exposed to hyperoxia.

They showed that hyperoxia-induced decrease of proliferation in A549 cells can be rescued by ATRA.

Further, the results revealed hyperoxia-induced upregulation of genes responsible for inflammation and inhibition of genes responsible for cellular proliferation. ATRA pretreatment is able to reverse some of these detrimental changes on gene level.

Major comments:

1. Trypan blue is staining dead cells. It is not clear to me how proliferation can be judge by Trypan blue staining. Please claarify. It would be better to stain for Ki67 and do flow cytometry.

Response: Thank you for your comment. We have edited the manuscript to clarify that we are excluding nonviable trypan blue cells from counting viable cell numbers over time. Furthermore, we separated Figure 1 into 1A and 1B and revised text accordingly for clarity of presentation. We added supplemental Figure S1 to include the effect of hyperoxia on Ki67 transcript expression. We acknowledge that we do not have protein and flow cytometry data for more rigorous assessment of cell cycle and have included this as a limitation in the discussion. 

2. Detection of some regulated genes of the Wnt and NF-kB pathway by RT-qPCR would underline and strengthen the RNA-seq results, because validation of the RNA-seq results is lacking.

Response: While we recognize that RT-qPCR experiments will complement our analysis, we are confident in the validity and robustness of our RNAseq dataset, as reported recently in this editorial piece: https://doi.org/10.1016/j.bioflm.2021.100043

3. Is ATRA toxic at higher doses, alone or in combination with hyperoxia?

Response: The optimal concentration range of ATRA to perform our experiments was selected based on prior studies (DOI:10.1371/journal.pone.0140343). While it is possible that ATRA is toxic at higher doses, alone or in combination with hyperoxia, our current study was not intended to determine toxicity at high concentrations. However, we have revised results and discussion section to indicate that ATRA exerts concentration-dependent effects on proliferation, survival, and differentiation and added pertinent citations.

4. The study design does not really allow a strong conclusion in the context of BPD pathophysiology. This limitation should be emphasized.

Response: We thank the reviewer for this comment. We have now edited the discussion to emphasize this limitation.

Minor comments

1. A549 cells can differentiate in the cell culture if they are kept to long and in higher passages ( Cooper JR, Abdullatif MB, Burnett EC, Kempsell KE, Conforti F, Tolley H, Collins JE, Davies DE. Long Term Culture of the A549 Cancer Cell Line Promotes Multilamellar Body Formation and Differentiation towards an Alveolar Type II Pneumocyte Phenotype. PLoS One. 2016 Oct 28;11(10):e0164438. doi: 10.1371/journal.pone.0164438. PMID: 27792742; PMCID: PMC5085087.). How long were the A549 cells in culture? What was the passage?

Response: We acknowledge the reviewer’s concern and have cited the above reference. Cells were used within a range of two passages for all experiments conducted, after being thawed from liquid nitrogen and cultured for ~3-4 days prior to being used in the experiment. We have also added Figure S2 to show transcript number for surfactant proteins and revised manuscript accordingly to indicate lack of evidence of differentiation in short term culture and differentiation potential of ATRA 

2. Please name the devices and companies in the materials and methods section.

Response: Following the reviewer’s recommendation, we have now included the missing names of devices and companies in the methods.

3. Were biological and/or technical triplicates performed? Please also name the n-number in the figures.

Response: Yes, each experiment was conducted in triplicate (n=3) with 3 technical replicates. This has been added to the figure legends to reflect the same.

4. Please check again for spelling and spaces.

Response: We have revised the manuscript for grammar and spelling.

5. Please be consistent in using hyperoxia-exposed, hyperoxia-treated, hyperoxia-challenged.

Response: We thank the reviewer for the suggestion. We have used hyperoxia-exposed throughout the revised manuscript for consistency.

6. The figures are blurry. Please improve the quality.

Response: Thank you. We have improved the quality of all images. We uploaded highest quality images and can work with editorial staff to ensure conversion of high-resolution images

Reviewer #2: 

Dear Authors, while the transcriptome analysis is well executed, the cellular proliferation assay showed in figure 1 should be revised.In particular the figure partially contradicts the statement at line 143 "Pretreatment with ATRA prior to hyperoxia exposure restored cellular proliferation after 48h" since the only statistical significant result at 48 hours regards the normoxia group treated wiTH 10-5 mM ATRA.

Response: Thank you for your comment. We have revised the results section to correct the error and separated Figure 1A and 1B for clarity of presentation of statistical data.

The graphical representation of the proliferation assay could be improved, and the different comparisons between groups and times is not well specified. From a statistical point of view the proper test for this assay should be a two way ANOVA, since the 2 variables (hyperoxia time and ATRA concentration) and the results representation would probably be more clear using a linear representation of the different treatments over time.

Response: We thank the reviewer for this comment. As indicated in the methods, our statistical analysis was indeed a two-way ANOVA, using hyperoxia induction and ATRA concentration as the two variables. We chose this analysis since our main goal was to determine whether ATRA treatment reduced hyperoxia responses, which is represented in Figure 1. Furthermore, we have revised the results section to correct the error and separated Figure 1A and 1B for clarity of presentation of statistical data.

---

## [Editor Report · Decision Letter 1]

26 Jul 2022

All trans-retinoic acid modulates hyperoxia-induced suppression of NF-kB-dependent Wnt signaling in alveolar A549 epithelial cells

PONE-D-22-12782R1

Dear Dr. Chroneos,

We’re pleased to inform you that your manuscript has been judged scientifically suitable for publication and will be formally accepted for publication once it meets all outstanding technical requirements.

Kind regards,

Saverio Bellusci

Academic Editor

PLOS ONE
---

## [Editor Report · Acceptance letter]

1 Aug 2022

PONE-D-22-12782R1 

All trans-retinoic acid modulates hyperoxia-induced suppression of NF-kB-dependent Wnt signaling in alveolar A549 epithelial cells 

Dear Dr. Chroneos:

I'm pleased to inform you that your manuscript has been deemed suitable for publication in PLOS ONE. Congratulations! Your manuscript is now with our production department. 

Kind regards, 

on behalf of

Dr. Saverio Bellusci 

Academic Editor

PLOS ONE